# Plasticity of *Plantago lanceolata* L. in Adaptation to Extreme Environmental Conditions

**DOI:** 10.3390/ijms241713605

**Published:** 2023-09-02

**Authors:** Zbigniew Miszalski, Paweł Kaszycki, Marta Śliwa-Cebula, Adriana Kaczmarczyk, Miron Gieniec, Paulina Supel, Andrzej Kornaś

**Affiliations:** 1The W. Szafer Institute of Botany, Polish Academy of Sciences, Lubicz 46, 31-512 Kraków, Poland; z.miszalski@botany.pl (Z.M.); a.kaczmarczyk@botany.pl (A.K.); m.gieniec@botany.pl (M.G.); 2Department of Plant Biology and Biotechnology, Faculty of Biotechnology and Horticulture, University of Agriculture in Krakow, Al. Mickiewicza 21, 31-120 Kraków, Poland; pawel.kaszycki@urk.edu.pl (P.K.); m.sliwacebula@gmail.com (M.Ś.-C.); paulina.supel@urk.edu.pl (P.S.); 3Institute of Biology and Earth Sciences, Pedagogical University of Krakow, Podchorążych 2, 30-084 Kraków, Poland

**Keywords:** Błędów Desert, ^13^C and ^15^N discrimination, chlorophyll *a* fluorescence, mycorrhiza, plant anatomy

## Abstract

This study aimed at characterizing some adaptive changes in *Plantago lanceolata* L. exposed to harsh conditions of a desert-like environment generating physiological stress of limited water availability and exposure to strong light. It was clearly shown that the plants were capable of adapting their root system and vascular tissues to enable efficient vegetative performance. Soil analyses, as well as nitrogen isotope discrimination data show that *P. lanceolata* leaves in a desert-like environment had better access to nitrogen (nitrite/nitrate) and were able to fix it efficiently, as compared to the plants growing in the surrounding forest. The arbuscular mycorrhiza was also shown to be well-developed, and this was accompanied by higher bacterial frequency in the root zone, which might further stimulate plant growth. A closer look at the nitrogen content and leaf veins with a higher number of vessels and a greater vessel diameter made it possible to define the changes developed by the plants populating sandy habitats as compared with the vegetation sites located in the nearby forest. A determination of the photosynthesis parameters indicates that the photochemical apparatus in *P. lanceolata* inhabiting the desert areas adapted slightly to the desert-like environment and the time of day, with some changes of the reaction center (RC) size (photosystem II, PSII), while the plants’ photochemical activity was at a similar level. No differences between the two groups of plants were observed in the dissipation of light energy. The exposure of plants to harsh conditions of a desert-like environment increased the water use efficiency (WUE) value in parallel with possible stimulation of the *β*-carboxylation pathway.

## 1. Introduction

Some plant species show high plasticity and can occupy a broad spectrum of different ecological niches. The plasticity of photosynthesis in the leaf blade, as well as the vascular tissues, may demonstrate an acclimation value for the plant performance under stressful conditions [1]. Light intensity and quality are among the most important factors affecting a plant’s response to stress [2]. The most intensive photosynthesis takes place in the leaf mesophyll cells; however, other tissues, such as the leaf veins or stems, are also often equipped with chloroplasts. The leaf vascular system, also called the “veins”, is typically encircled by bundle sheath (BS) cells containing chloroplasts, and green cells can be also found within the phloem and xylem inside the vascular tissue [3,4,5]. Green cells located within the vascular system are able to absorb light, and this energy can be used for NADP^+^ reduction and ATP production. In C_4_ plants, the cells that build leaf veins show clear structural and ultrastructural differences compared to mesophyll, and they play an important role in CO_2_ assimilation (Kranz anatomy); however, their role in C_3_ plants has not yet been well described. The functions of the vascular system are linked to mechanical support, as well as transport and signaling, and they are considered important for integrating the whole plant response, which means communication among the leaves, stems and roots, to the stress circumstances [3,4,5]. Green cells found within the vascular system are involved in the transport of sugars to the phloem, known as “phloem loading” and are possibly involved in the long-distance signaling mediated by H_2_O_2_ and phytohormones such as NO, and in processes which enable the plant to acclimatize to changes in the environment at the whole-plant level [6]. Some publications indicate clear differences in the ultrastructure of chloroplasts between the mesophyll and BS, which occurs in parallel with the lower activity of the photosystem II (PSII) and a lower PSII/PSI ratio in the chloroplasts from the veins [6,7]. The high photochemical efficiency of PSII is usually found in the leaves of high light-acclimated plants, while a relatively high PSI content is characteristic of plants grown in low light, photosynthetically active radiation (PAR) [8].

The vascular parenchymal cells are rarely considered in terms of their role in photosynthesis; however, these tissues can be supplied with photorespiratory and respiratory CO_2_ or carbon originating from the decarboxylation processes of malate transported from other tissues in the root, in a process similar to the one described for C_4_ plants [9,10]. Despite the high concentration of CO_2_ and the expected lower level of O_2_ usually found in more heterotrophic tissues, in green cells located within the veins, the photosynthetic activity is lower than in the surrounding mesophyll cells. This could result also from the fact that the CO_2_-rich environment impedes the photochemical activity, possibly through acidification of the protoplasm, which is followed by a reduction in the efficiency of heat dissipation, described by Paul and Foyer [11] as an effect of acclimation. As shown in experiments with the midribs and stems, cells located close to the veins are equipped with carboxylating enzymes and may use some amounts of CO_2_ for photosynthesis [4,5,12,13]. We expect that stems and vascular tissues (midribs) can use, at least in part, *β*-carboxylation, and that the “new” CO_2_ (not resulting from respiration) can be fixed and can participate in biomass building. In addition, the transcripts for both *pepc1* and *nadpme1* genes, responsible for *β*-carboxylation (C_4_ and crassulacean acid metabolism, CAM photosynthesis), were found in the veins of C_3_ and CAM plants [13,14]. This was also supported by a ^13^C discrimination factor indicating the possible involvement of phosphoenolpyruvate carboxylase (PEPC) in the fixation of CO_2_ within the veins of many plants [7]; however, according to Pfanz et al. [15] and Kocurek et al. [5], the vascular system contribution to the total productivity is rather low. In the midribs of *Mesembryanthemum crystallinum* leaves (in the C_3_ state), when compared to the leaf mesophyll, significantly lower rbcL protein abundance (a large RubisCO subunit) was observed, which indicates that the CO_2_ fixation was not intensive. Interestingly, in the midrib tissues of *Plantago media* leaves, the photosynthetic electron transport (PET) [7] is somehow even more efficient than in the lamina, and it is not disturbed under high light conditions. In the midribs of this plant, there is high electron transport rate (ETR) efficiency, and the resulting energy can support veinal tissues in their transport function. PEPC, which is responsible for *β*-carboxylation, also plays a crucial role in a variety of important non-photosynthetic anaplerotic functions [16,17,18], including nitrogen fixation, known as a process necessary for synthetizing proteins, nucleic acids and many other biomolecules [19]; thus, PEPC creates, at least partially, carbon–nitrogen interactions, and some PEPC isoforms are more specifically expressed in the roots [20]. As reported by Hibberd and Quick [9], and Brown et al. [21], PEPC localized in the root tissues is responsible for significant CO_2_ fixation in tobacco and *Arabidopsis* plants. The absence of PEPC leads to suppressed ammonium assimilation [22]. *Arabidopsis* mutants with some impaired PEPC isoforms could fix only 10% of the available N [18]. For the reasons above, one can expect that veins with their BS green cells and active PEPC may play an important role in nitrogen and CO_2_ fixation processes.

Some plants, such as *M. crystallinum*, *Clusia* sp., *P. media* or *P. lanceolata*, develop very clear veins surrounded by several parenchyma layers. These species often show resistance to different environmental factors and drought stress, and it is expected that clear veins (midribs) play a role in adaptation of plants to stress. Herein, the analyses of photosynthesis-related properties were applied to answer the question if they play a significant role in acclimation to this kind of stress. Then, the question arises if there are any anatomical differences in the midrib anatomy between the plants grown in the forest and desert environments. The aim of this study was to detect any possible changes in the anatomy and in the course of photosynthesis that could be associated with *P. lanceolata* growing under harsh desert environmental conditions with strong light and limited access to water resources. We focused on the plant organs responsible for water and nutrient transportation and the availability of nitrogen in the analyzed locations. Note that this plant is known as a species that is able to grow on sandy and clayey sites in the presence of environmental contaminants [23] and can effectively develop its photosynthesizing organs. Are the midribs of *P. lanceolata* and the microbial frequency within the root zone growing in extremely low water supply areas similar to those growing in the forest? Here, it has to be emphasized that the research on the adaptational potential of plants inhabiting depleted ecosystems has recently become an important issue in the context of global climate change that leads to expected progressive degradation of large areas.

## 2. Results

### 2.1. Leaf Morphology and Anatomy

A comparison of the morphological features of *P. lanceolata* showed that the roots of plants grown in the Błędów Desert (BD) and in the surrounding forest (YF) locations were fundamentally different (Figure 1). The plants collected at BD, compared to those growing in YF, displayed better developed root systems. The plant shoots also showed morphological differences. The leaves of the BD plants were longer (12.6 cm; SD ± 4.36) compared to those of the YF site (8.45 cm; SD ± 3.57); however, the difference is not significant. Likewise, the leaves of the BD (1.31 cm SD ± 0.38) plants were slightly broader compared to the ones collected at the YF site (1.22 cm; SD ± 0.38); however, the difference is not significant, either. Similarly, the length/width ratio of the leaves of the BD plants (9.9 cm; SD ± 3.3) was slightly higher compared to the leaves of the YF site (7.1 cm; SD ± 2.8); here, the difference is also not significant.

Figure 2 shows the leaf vascular bundle cross-sections of *P. lanceolata*, which prove that the vascular tissue was better developed for plants grown in BD and was especially pronounced within the area of the main leaf vascular tissues. An analysis of the representative vascular bundle morphology showed that the average number of vessels was higher (n = 61) and the diameter of the vessels was larger (8.74 µm ± 2.77) in the plants growing in BD in comparison to the plants growing in the YF locations (n = 49; 6.80 µm ± 2.98). In addition to this, we have found that the N/C ratio in the plants from BD was higher than that obtained for the YF locations (the lamina was about 1.98 times higher, and the leaf bundle was 1.55 times higher—it was a significant increase).

### 2.2. Microbial Colonization and Mycorrhiza 

The soil microbial population analyses shown in Figure 3 reveal that the rhizosphere of the plants growing in the BD locations was colonized by a statistically significant higher number of bacteria, whose frequency was approx. 9 times greater relative to the YF locations (5.01 ± 1.90 × 10^7^ and 5.61 ± 4.10 × 10^6^ CFU/g d.m., respectively). At the same time, the number of yeast within the root zone was similar for both cases (1.07 ± 0.53 × 10^5^ CFU/g d.m. for YF and 1.50 ± 0.22 × 10^5^ CFU/g d.m. for BD). 

The mycorrhiza was much better developed for the BD locations than in the forest area. Although a morphological microscopic evaluation of the root fragments obtained at both sites did not reveal significant differences (Figure 4), a detailed analysis of arbuscular mycorrhiza colonization (Table 1), based on multiple sampling and taking into account the enhanced development of the root system at BD, shows considerable changes in favor of plants growing in the desert. 

The intensity of mycorrhizal colonization (M%, m%) was stronger by 14%, and the arbuscule abundance (A%, a%) was significantly increased (6.0 and to 5.2 times higher for A% and a%, respectively) in the mycorrhizal parts of the *P. lanceolata* root fragments obtained from BD compared to the surrounding forest (YF).

### 2.3. The Leaf Signature: ^13^C and ^15^N Discrimination

The carbon isotope ratios clearly indicate that the desert environment generated multiple physiological stresses. For that reason, the *P. lanceolata* plants probably increased their water use efficiency (WUE) during photosynthesis, which was manifested by lower discrimination and a less negative δ^13^C value (Figure 5) found for the lamina, leaf bundle and bigger roots, as well as the smaller roots. Interestingly, δ^13^C differed by about 2‰ for the lamina, leaf bundle and bigger roots for both habitats, and all the values were less negative as compared to the lamina.

Discrimination against ^15^N in all the tested tissues was higher (less negative values) for the samples obtained from the BD locations. In the samples from the YF locations, the highest discrimination was observed for the main roots; however, the small roots (lateral) showed no discrimination (positive values).

### 2.4. Analysis of the Content of Various Forms of Nitrogen in the Soil

The content of nitrogen in the form of N-NO_2_ was similar, although it showed a slight tendency to be higher in the desert location. In the case of N-NO_3_, we noted a statistically significant difference (Figure 6). The opposite relationship was found for N-NH_4_.

### 2.5. Analyses of Chlorophyll a Fluorescence

For the leaves analyzed in the Błędów Desert (BD) area, the photosynthesis parameters Fv/Fm (indicating oxidative stress; Figure 7A), ET_0_/RC (determination of the electron transport velocity per PSII reaction center; Figure 7B) and TR_0_/RC (energy capture by a single reaction center; Figure 7C) were slightly lower than in the forest (YF) locations, and this tendency could be observed at each investigation time. At the same time, the parameter DI_0_/RC, representing the total dissipation (heat or fluorescence) of the energy not captured by a single reaction center, was not significantly changed (Figure 7D). This could imply that the components responsible for the DI_0_/RC ratio adapted to the new conditions very quickly, proving statistically significant changes at all the measurement hours, except for 11:00, and, taken together, indicating photoinhibition; thus, these parameters were rather stable throughout the course of the day, which suggests that this phenomenon may have resulted from the lack of differences in the structure of some components of the electron transport chain (PSI and PSII). The findings above support the idea that the tested desert-growing plants undergo physiological stress associated with drought and excessive exposure to sunlight.

Surprisingly, in the tested desert-growing plants, this parameter was not significantly higher in comparison to the plants growing in the surrounding forest. All the collected data indicate that the photochemical apparatus in the plants populating desert areas adapted slightly to strong light and drought by reducing the reactive center (RC) size while keeping their photochemical activity at a similar level.

## 3. Discussion

*P. lanceolata* L. (Plantaginaceae) is known as a plant that usually occupies different locations, often found on permanent and non-permanent grasslands. It can grow in various soils, including sandy sites. Its very rapid growth and deep rooting in the soil results in high drought tolerance and efficient uptake of valuable nutrients from the deep soil layers. We have performed our research on plants occupying either sandy soils of the BD or the young forest (YF), inhabited mostly by *Pinus sylvestris* and located in the area surrounding the desert (Figure 1). The leaves of both plant groups differed strongly; those of the BD area were slightly wider than those growing in the YF locations (Figure 1 and Figure 2).

We have also compared the root systems and found that the plants growing in desert sands produced much bigger and denser root systems with well-developed parts of the lateral roots. For optimal root development, most plants require enhanced symbiosis with rhizospheric microorganisms and, in particular, with mycorrhizal fungi.

The nine-fold increase of *P. lanceolata* root-zone bacterial frequency, as observed for the samples collected at the BD locations relative to the YF ones, indicate that the desert sandy soil, although low in humus, still can provide favorable conditions for the proliferation of bacteria, provided these microorganisms grow under a canopy of plants forming “vegetation islands”. Similar observations were reported by Bachar et al. [24], who indicated a substantial increase in the bacterial abundance within the areas of “resource islands” with well-developed vegetation and pointed out the high changeability of bacterial consortia depending on the island patches and inter-patched arid places.

It is worth noting that in the bulk soil typical of arid or desert areas, a substantial reduction in the total bacterial biomass is usually observed. This is mainly due to nutrient limitation in poor-quality soils with water deficit caused by drought stress [25,26]. In this study, we focused on a comparison of culturable microbiota inhabiting the root zone of *P. lanceolata* growing in two distinct locations (BD and YF). Due to exposure to extremely variable climatic conditions, especially water scarcity, the sandy soil itself was poorly populated with microorganisms (within the range of 10^3^–10^4^ CFU/g d.m.). Microbial abundance analyses carried out by Köberl et al. [26] showed that Egyptian desert soil was populated with approx. 10^4^ CFU/ g d.m., while the colonization rate of agricultural soil was three orders of magnitude higher. In our work, the high vegetation density in the surrounding forest area made it impossible to collect bulk soil samples not penetrated by plant roots for reference microbial population analyses.

The well-developed root system in the BD area was most likely an important factor promoting microbial colonization by serving as a suitable habitat and leading to the higher abundance of rhizospheric bacteria. This, in turn, can be expected to stimulate plant growth in unfavorable desert conditions. Such a dynamic mutual interaction is known as the rhizosphere feedback loop and becomes particularly important for perturbed environments where plant root-associated beneficial microbes may contribute to plant health [25,27].

The mycorrhizal colonization of *P. lanceolata* roots was observed at both locations; however, for the BD-originating plants, it was significantly enhanced, resulting in both a heightened intensity of colonization (M%, m%) and an elevated arbuscule abundance (A%, a%). Since, in our observations, the roots of the plants growing in the desert were thicker, denser and spreading, they were prone to infection with mycorrhizal fungi. As emphasized by Alkobaisy [28] in a recent review, it is well proven that mycorrhiza propagation is very strongly influenced by the plant root system’s density. It is also known that in fertile soils, the intensity of mycorrhizal colonization is relatively lower, while the arbuscular fungal development within the root tissue is strictly controlled by the host. In turn, Vasar et al. [29] stressed the fact that enhanced development of arbuscular mycorrhiza is a particularly important phenomenon occurring in drought desert systems. Mycorrhizal fungi are known to help plants in the uptake of water and some nutrients. Fungi participating in arbuscular mycorrhiza are the most widespread plant symbionts, occurring in over 80% of terrestrial plants [30]. In the case of arbuscular mycorrhiza, the symbiotic fungi are localized in the cortex of plant roots where arbuscules are formed. A strong interaction with mycorrhizal fungi is a characteristic feature of plants that tend to increase their capacity for nutrients and water uptake (Figure 3).

It can be expected that a well-developed lateral root system equipped with mycorrhizal fungi can significantly reduce the drought stress to which these plants are exposed, and can be responsible for delivering the necessary amount of nutrients.

*P. lanceolata* is an obligate C_3_ plant with primary photosynthetic CO_2_ fixation via RubisCO; however, it is important to note that all the plants are also equipped with carboxylating PEPC, which is involved in a variety of non-photosynthetic anaplerotic roles, including nitrogen fixation [16,17,19]. Its carboxylating activity enabling catalysis of CO_2_ into oxaloacetic acid is well described in the stomata of C_4_ and CAM plants. In C_4_ plants, PEPC is responsible for CO_2_ fixation. It is the most important protein that accumulates carbon within the mesophyll cells, and after decarboxylation within the bundle sheath, CO_2_ is refixed with RubisCO (Kranz anatomy). Additionally, in the leaves of CAM plants, similar activity enables the accumulation of carbon during the nighttime. The activity of both carboxylases as a primary fixation step can be distinguished with the help of their features in discrimination of the ^13^C isotope, and some insight into biomass building can be provided by the analyses of the ^13^C/^12^C ratio. While RubisCO has a strong preference for the lighter isotope ^12^C over the heavier one ^13^C (described as δ^13^C), resulting in a discrimination of ^13^C at −29.0‰ at the enzyme level, PEPC is less discriminating at −5.7‰; however, during the overall photosynthetic process, this is overlaid by carbon isotope discrimination of ^13^CO_2_ diffusion, particularly via the stomata (−4.4‰). When the stomatal conductance is high, the discrimination of ^13^CO_2_ is lower (less negative δ^13^C values) than for the case when diffusion via the stomata is restricted, and along this same line, the stomatal conductance is inversely related to WUE. Decreasing the stomatal conductance results in lowering the leaf intercellular CO_2_ level (C_i_), which finally leads to an increased WUE [31,32,33]; thus, plants suffering from water deficit usually use water more efficiently when calculated as the amount of fixed CO_2_. The carbon isotope ratios clearly indicate that the desert environment generated physiological stress of limited availability of water. For that reason, *P. lanceolata* increased its WUE during photosynthesis, which was manifested by lower discrimination and a less negative δ^13^C value (Figure 5); δ^13^C differed by about 2.6–3.0‰ for the lamina, leaf bundle and main (bigger) roots in both habitats and only 0.89‰ for the lateral roots, whereas all the values were less negative compared to the lamina. Several factors are expected to be involved in contributing to these differences. In addition to WUE, an increased activity of PEPC, which is characterized by a lower ^13^C discrimination, especially in heterotrophic tissues, can reflect the lowering of the δ^13^C value for the case of stress-exposed plants [7,34,35]. In our former research testing *Clusia rosea*, we found the PEPC protein within the veins [13]. It can also be expected that vascular tissues in the roots play a role in nitrogen fixation processes, thus creating carbon–nitrogen interactions. Access to nitrogen is one of the most important factors enabling plants to grow, especially under conditions of stress exposure with a limited possibility of CO_2_ fixation. Even more interestingly, in experiments on *Clusia alata* plants, high light accelerated malate decarboxylation, providing more CO_2_ for photosynthesis [36].

In our experiments, we were able to show that the BD locations were richer in two nitrogen-containing anions (NO_2_^−^ and NO_3_^−^) when compared to YF, and as expected, the level of ammonium cations (NH_4_^+^) was lower. The latter value possibly reflects the amount of organic matter present in both soil samples, and the YF soil is richer in organic nitrogen and contains a high amount of organic matter (roots of many plants, bacteria, fungi). In the YF, locations many different plant species compete for nitrogen occurring in different forms, and in this way, they lower its availability; thus, plants growing in the BD locations are better supplied with nitrogen. In addition, while analyzing the ^15^N discrimination values (Figure 5), one can conclude that the BD plants are better supplied with nitrogen compared to YF. Note that the nitrogen isotope discrimination varies in accordance with the changes in the nitrogen supply, and the heavier isotope (^15^N) is “discriminated” against, causing a relatively greater fraction of the lighter isotope (^14^N) to be incorporated into the biomolecules [31,37].

A rich source of nitrogen in the form of NO_3_^−^ and NO_2_^−^, as shown in Figure 6, needs a physiological system ready for its absorption, reduction and incorporation into biomolecules. A large part of these processes take place in the vascular system of plants [38]. In our study, a greater difference in the ^15^N discrimination values (positive versus negative) between BD and YF, as observed for the lateral roots in comparison with the other tested tissues, can also support the suggestion above, indicating differences in the nitrogen supply from the soil for both compared locations. The most visible difference was observed for the lateral roots (Figure 6, positive versus negative values), indicating again a low amount of nitrogen in the soil and/or a high demand for nitrogen in these locations. Interestingly, only the plants growing in YF showed low WUE values and high photochemical activity (Figure 5 and Figure 7), while at the same time needing the highest amount of nitrogen necessary for synthesis of large amounts of the RubisCO protein. The less negative ^13^C discrimination values observed for the lateral roots could also indicate the high activity of PEPC in this tissue. Differences in the ^15^N discrimination between both locations, BD and YF, are also clearly visible when other tissues are compared, namely, the lamina, leaf bundle and main roots, which suggests that following the transportation of nitrogen to the leaf lamina and the reduction and incorporation of biomolecules, ^15^N discrimination is less important; thus, during the next steps of nitrogen assimilation inside the root and stem tissues, the plants use both isotopic forms (^14^N and ^15^N).

From these findings, we suggest that plants growing in the BD locations can take up more nitrogen than the YF plants. The higher availability of different nutrients also enabled other plants, such as *Salix arenaria*, to develop at very similar BD locations, as was shown in our previous research [39]. In the aforementioned study, it was found that this tree was capable of adapting most of its photosynthesis parameters to enable efficient vegetative performance as a robust pioneer organism. *P. lanceolata* can also be expected to function as a nurse plant, providing conditions that promote the growth of other plants and stimulate proliferation of microorganisms. The plant’s ability to take up, transport and reduce nitrogen appears to be an important factor contributing to the nursing process.

As shown above, the desert environment generated physiological stress of limited water availability and increased the plants’ WUE during photosynthesis. The effect of unfavorable conditions is usually well reflected by PS II efficiency [40,41]. All the collected data indicate that the photochemical apparatus in *P. lanceolata* inhabiting desert areas (Figure 7) acclimated slightly to strong light and drought, which involved a photoinhibition mechanism and the reduction of the reaction center (RC) size of PSII, while maintaining the plant’s photochemical activity at a similar level. Similar changes of this parameter for the BD and YF plants during the daily course allow us to speculate that this system is rather flexible and can easily acclimate throughout the day to the light conditions. This, in turn, leads to decreased values of the following parameters: the amount of energy absorbed and the rate of electron transport. A decreased potential PS II yield typically results from the elevated energy amount dissipated as heat and fluorescence; however, excess of this parameter changing during the daily course was similar for plants growing in both the BD and YF locations.

## 4. Materials and Methods

### 4.1. Study Sites and Plant Material

The Błędów Desert (“Pustynia Błędowska”) is a complex of two sand habitats, that is, inland dunes and stenothermic grasslands, covering a total area of approximately 33 km^2^. It has been classified under the European Framework Programme, which is dedicated to networks of landscape protection areas, as NATURA 2000—Area Code PLH-120014 [42]. The desert is located in the northwest part of the so-called Jura Krakowsko-Częstochowska (Polish Jurassic Highland or Cracow-Częstochowa Jura Chain), on the borders of the Śląska (Silesian) and Olkuska Highlands (50°21′23.4″ N 19°30′56.3″ E) [43]. Among the most important species are sand ryegrass (also known as sea lyme grass or beach wildrye, *Leymus arenarius* (L.) Hochst. (syn. *Elymus arenarius* L.)), sharp-leaf willow (*Salix acutifolia* Willd.) and creeping willow (*Salix repens* subsp. *arenaria* (L.) Hiit.) [44]. Although a relatively less-dominant species, *Plantago lanceolata L.* can be found both at numerous sandy sites and in the forest surrounding the desert. The Błędów Desert in the southern part of Poland is formed of fluvioglacial sand (average thickness of 40 m) with an admixture of gravel from the Pleistocene. The mean annual air temperature in the study area is 7 °C and the annual precipitation is 650 to 750 mm [45]. The Błędów Desert is surrounded in the north and south by forest complexes; hence, characteristic dune border ridges have been formed on the verge between the flat sandy surface and the forest [46]. *P. lanceolata* usually grows on light, well-aerated soils; grazing, sandy grasslands; or black earth.

### 4.2. Leaf Morphology Parameters

Cross-sectional hand-cut fragments of *P. lanceolata* leaves were observed in distilled water using a Nikon ECLIPSE Ni light (Nikon, Tokyo, Japan) equipped with a microscope camera Digital Sight series DS-Fi1c and NIS Imaging software, Nikon version 4.11.

### 4.3. Bacterial Frequency Determination and Mycorrhizal Colonization Assessment

Analyses of the microbial frequency within the root zone of *P. lanceolata* plants inhabiting the desert and forest areas were performed after carefully pulling out the root systems, followed by shaking off and collecting the soil material. Both bacteria and microscopic fungi (mainly yeast) were characterized in aqueous soil extracts with a standard Koch surface-plating method, using Petri dishes that contained solidified media: 2.5% enriched agar or 6.5% Sabouraud medium (BioMaxima, Poland), respectively. The well-developed colonies were evaluated macroscopically and then counted after 3 to 4 days of incubation at room temperature. To calculate the biomass density, the cell frequencies were expressed as CFU (colony-forming units) per g d.w. of the original soil samples [47].

A mycorrhiza assessment was carried out by staining the fungal structures inside the root via the modified method of Phillips and Hayman [48] described by Ważny et al. [49]. The roots were washed in tap water, cleaned in 10% KOH for 24 h, washed again, acidified in 5% lactic acid for 2 h and stained with 0.01% aniline blue in pure lactic acid for 24 h at room temperature. The stained roots were stored in pure lactic acid, cut into 1 cm pieces and mounted in glycerol on microscopic slides. In the quantitative analyses, at least three plants were randomly selected and 30 root pieces per plant were examined.

The mycorrhizal frequency (F%), intensity of mycorrhizal colonization (M%), absolute mycorrhizal colonization (m%), arbuscule abundance in the root system (A%) and absolute arbuscular richness (a%) were assessed. To calculate the parameters, %F, %M, %m, %a and %A, the following formulae were used (https://www2.dijon.inrae.fr/mychintec/Protocole/protoframe.html, accessed on 15 July 2023) from Trouvelot et al. [50]:
Frequency of mycorrhiza in the root system
$$F\%=\frac{nb\;of\;fragments\;myco}{total\;nb}\cdot 100$$

Intensity of the mycorrhizal colonization in the root system$$M\%=\frac{\left(95n5+70n4+30n3+5n2+n1\right)}{total\;nb}$$where n5 = number of fragments rated 5 and n4 = number of fragments, i.e., 4, etc.


Intensity of the mycorrhizal colonization in the root fragments

$$m\%=M\cdot \frac{total\;nb}{nb\;of\;fragments\;myco}$$



Arbuscule abundance in the mycorrhizal parts of the root fragments$$a\%=\frac{\left(100mA3+50mA2+10mA1\right)}{100}$$where mA3, mA2 and mA1 are the % of m, rated A3, A2 and A1, respectively, with

mA3 = $\frac{\left(95n5A3+70n4A3+30n3A3+5n2A3+n1A3\right)}{\mathrm{nb}\;\mathrm{myco}}\cdot \frac{100}{m}$ and the same for A2 and A1.


Arbuscule abundance in the root system

$$A\%=a\cdot \frac{M}{100}$$



### 4.4. The Content of Carbon ^13^C and Nitrogen ^15^N Isotopes

Prior to the analyses of the carbon and nitrogen isotopes content, freeze-dried leaf material was homogenized in an agate mortar until a fine-grained powder was obtained. A determination of the carbon isotope ratio of *δ*^13^C and *δ*^15^N was performed with the continuous flow technique. The samples were weighed and then burned off in a furnace at 1020 °C. Next, they were transported within a helium stream through the Con-Flo IV Interface to the mass spectrometer to enable a determination of the ratio of the particular carbon isotope contents. The calculations were computed while employing internal standards USGS 40, USGS 41 and IAEA 600 [51].

### 4.5. The Content of Various Forms of Nitrogen in the Soil

Air-dried and sieved (2 mm mesh) soil samples were shaken in water for 1 h (1:10, *w*:*v*) using a 358S shaker (Elan, Warsaw, Poland). A DX-100 ion chromatograph (Dionex, Sunnyvale, CA, USA) was used to determine the N-NH_4_, and the ICS-1100 ion chromatograph (Dionex, USA) was used to determine the N-NO_2_ and N-NO_3_.

### 4.6. Chl a Fluorescence Measurements

Photochemical parameter analyses were carried out using a Handy Plant Efficiency Analyzer spectrofluorometer (PEA, Hansatech Instruments Ltd., Norfolk, UK) according to the manufacturer’s manual. The fluorescence parameters of the plant leaves were measured on the upper side of the leaf blade. Clips with a 4 mm diameter hole were clamped on the leaf to be tested for their adaptation to the dark after 20 min. Radiation of 3 mmol (quantum) m^−2^ s^−1^ was used for the excitation of chlorophyll fluorescence. The measurements were taken four times at 8:00, 11:00, 14:00 and 17:30 h. Energy flow through PSII was evaluated on the basis of the flow parameters: ET_0_/RC—rate of electron transfer through the active reaction center (RC); TR_0_/RC—energy trapping of one active reaction center; DI_0_/RC—total energy dissipation, not trapped by RC. The analyses were conducted under favorable atmospheric conditions, during warm and sunny days in July–August (maximum temperature of approximately 25 °C).

### 4.7. Statistical Analysis

The statistical analysis was performed with the Statistica 13.3 package (TIBCO Software Inc., Palo Alto, CA, USA). The significance between the means for the fluorescence chl*a* parameters was analyzed by Duncan’s test and verified at *p* ≤ 0.05. The single-point chlorophyll fluorescence data represent the mean of 20 measurements per treatment grouped in four replications ± standard deviation (SD). The significance between the means for the leaves’ parameters were analyzed by *t*-test and verified at *p* ≤ 0.05.

The values of δ^13^C and δ^13^N and the mycorrhizal frequency were analyzed by a *t*-test at *p* ≤ 0.05. For the isotope analyses, the data represent the means of 3 independent measurements. The arbuscular mycorrhiza was assessed using 90 root specimens obtained from plants growing at each site (forest, YF or desert, BD). The rhizospheric bacterial and yeast populations were determined for at least four replicate samples collected from either the YF or BD locations. The frequencies of the microorganisms were given as mean values with standard deviation.

## 5. Conclusions

*P. lanceolata* plants growing in desert-like, sandy areas demonstrated an elevated rhizospheric bacterial abundance and additional access to a relatively rich source of nitrogen. All this enables them to produce large amounts of proteins in the form of RubisCO, among others, which are necessary for intense CO_2_ fixation under unfavorable conditions of drought stress exposure.

Taken together, our data illustrate that *P. lanceolata* may be considered as a pioneer organism capable of populating degraded areas with poor and arid soils. The plants can grow without large photochemical alterations, while revealing adaptational mechanisms enabling them to develop under harsh conditions, and as a result of this, they provide better environments for other plants and soil microorganisms. This is made possible as a consequence of possessing an effective mechanism enabling the use of nitrogen resources in such areas. Due to the plants’ ability to form large biomass, they may help to increase the content of humic substances in the soil. At the same time, this species may play an important role in slowing down the stepping process and decrease the susceptibility to erosion of sites devastated and over-exploited anthropogenically.

## Figures and Tables

**Figure 1 ijms-24-13605-f001:**
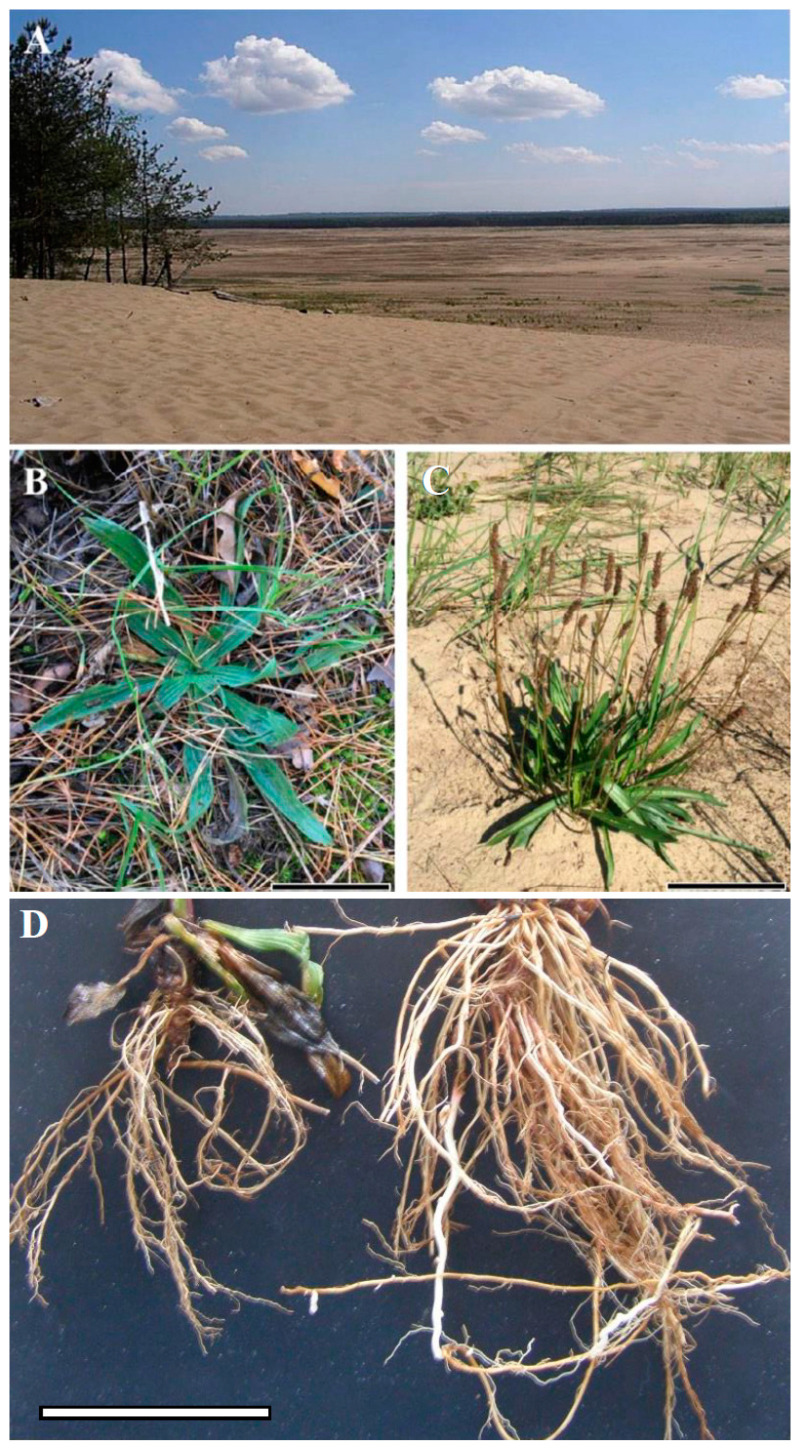
View of the Błędów Desert “Pustynia Błędowska” and the surrounding forest (**A**); morphology of *Plantago lanceolata* L. plants growing in the forest and desert (**B**,**C**); root systems of *P. lanceolata* L. plants growing in the forest (left side) and desert (right side) (**D**); scale bar: 5 cm.

**Figure 2 ijms-24-13605-f002:**
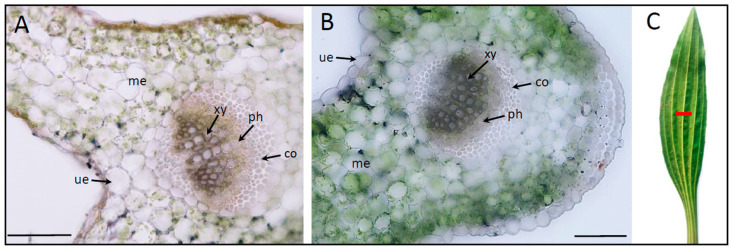
Anatomy of leaves of *Plantago lanceolata* L. plants collected from the surrounding forest (**A**) and in the Błędów Desert “Pustynia Błędowska” (**B**); diameter of the vessels in vascular tissues. The cross-section of the central part of leaves was observed using white light (**C**); scale bar: 100 μm; co—collenchyma; le—lower epidermis; me—mesophyll; ph—phloem; ue—upper epidermis; xy—xylem.

**Figure 3 ijms-24-13605-f003:**
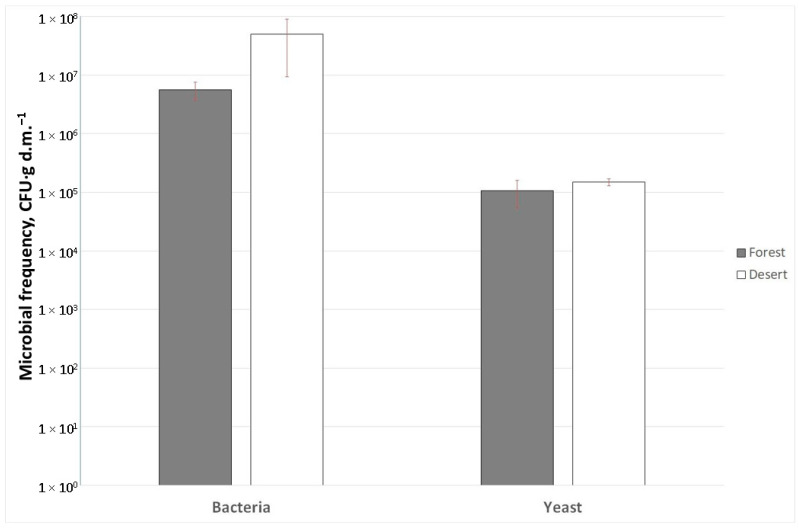
Bacterial and microscopic fungi (mainly yeast) occurrence determined as an averaged population frequency (CFU·g d.m.^−1^) in the soil of the root zone of *Plantago lanceolata* L. collected from the surrounding forest (dark bars) and desert (light bars) locations.

**Figure 4 ijms-24-13605-f004:**
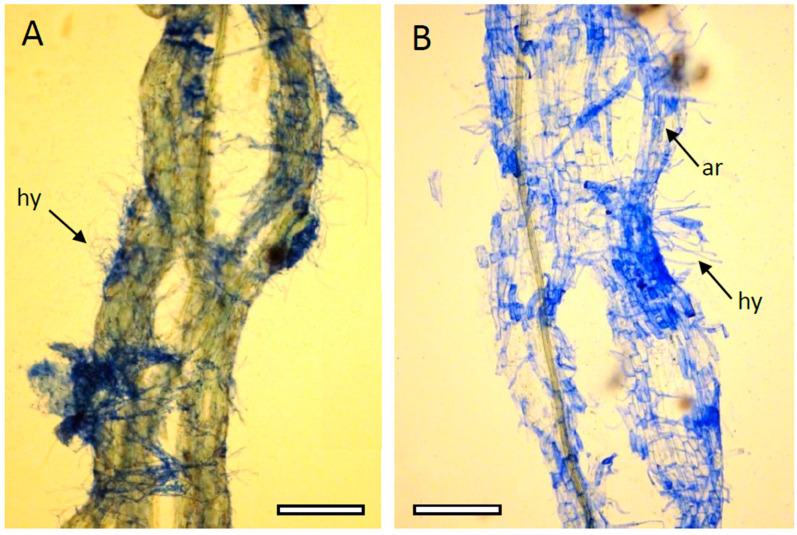
Arbuscular mycorrhiza in the roots of *Plantago lanceolata* L. plants growing in the surrounding forest (**A**) and in the Błędów Desert “Pustynia Błędowska” (**B**); ar—arbuscule; hy—hyphae; scale bar: 0.5 mm.

**Figure 5 ijms-24-13605-f005:**
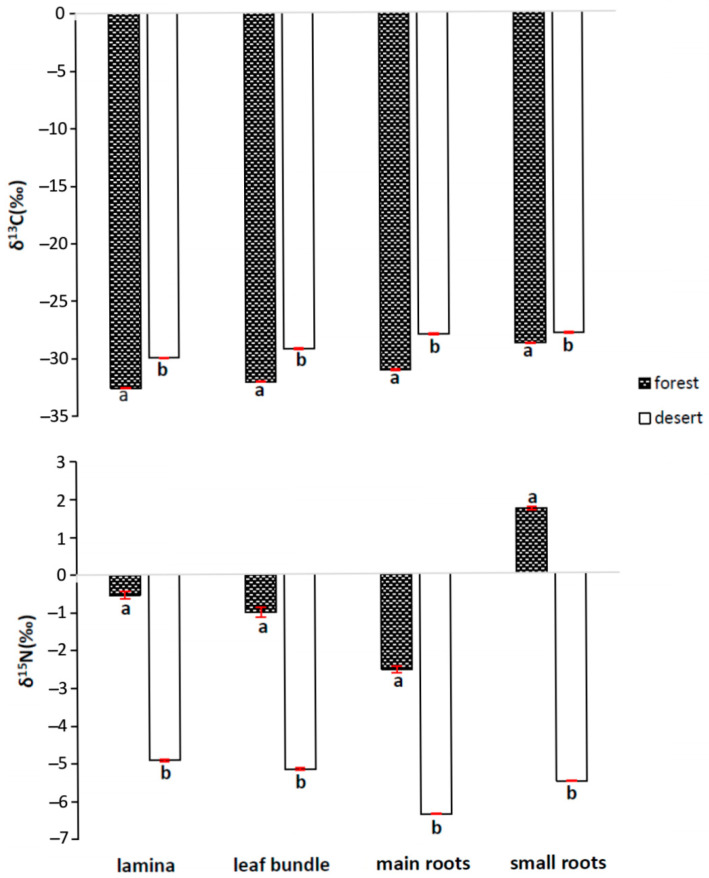
Values of δ^13^C and δ^15^N in lamina, leaf bundles and roots of *Plantago lanceolata* L. plants growing in the surrounding forest and in the Błędów Desert “Pustynia Błędowska”. The means followed by the same letters do not differ significantly according to the *t*-test at *p* ≤ 0.05 within the data analyzed for each parameter separately. n = 3; whiskers represent standard errors.

**Figure 6 ijms-24-13605-f006:**
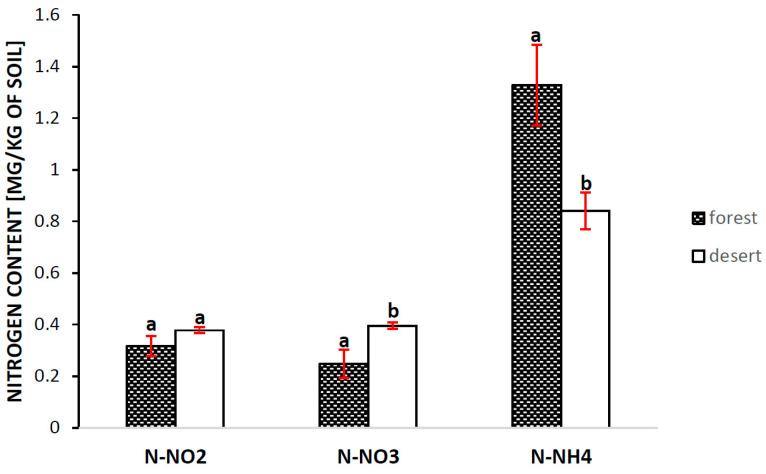
The content of various forms of nitrogen in the soil collected from the surrounding forest and the Błędów Desert “Pustynia Błędowska” where *Plantago lanceolata* L. plants were growing. The means followed by the same letters do not differ significantly according to the *t*-test at *p* ≤ 0.05 within the data analyzed for each parameter separately. n = 3; whiskers represent standard errors.

**Figure 7 ijms-24-13605-f007:**
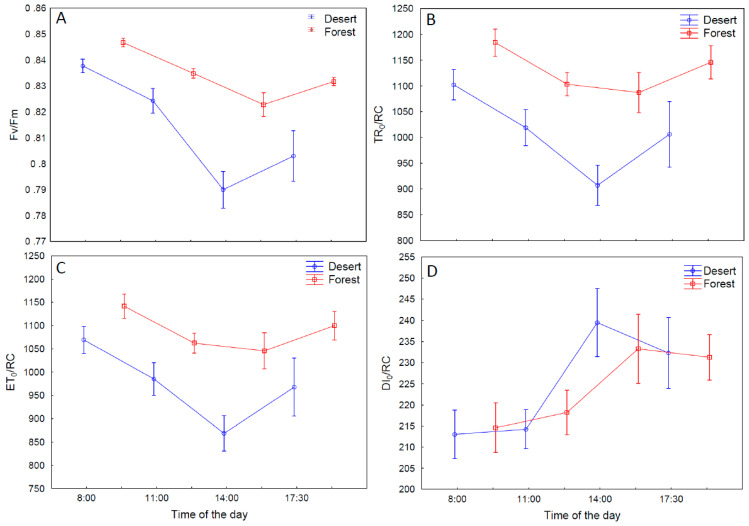
Maximum quantum yield of PSII (F_V_/F_M_) (**A**); trapping per active reaction center (TR_0_/RC) (**B**); electron transport per active reaction center (ET_0_/RC) (**C**) and dissipation per active reaction center (DI_0_/RC) (**D**), measured for *Plantago lanceolata* L. plants growing in the surrounding forest and in the Błędów Desert “Pustynia Błędowska”. n = 20; whiskers represent standard errors.

**Table 1 ijms-24-13605-t001:** Mycorrhizal colonization parameters calculated for *Plantago lanceolata* L. growing in the Błędów Desert and in the surrounding forest. F%, the frequency of mycorrhiza; M%, intensity of the mycorrhizal colonization in the root system; m%, intensity of the mycorrhizal colonization in mycorrhizal parts of root fragments; A%, arbuscule abundance in the root system; a%, arbuscule abundance in mycorrhizal parts of root fragments. Different letters indicate statistically significant differences at *p* ≤ 0.05 by a *t*-test with the data analyzed for each parameter separately; n = s90.

	Habitat	
	Forest	Desert
F%	100 ± 0.0	100 ± 0.0
M%	57.9 ± 3.8 a	66.1 ± 2.3 b
m%	57.9 ± 3.8 a	66.1 ± 2.3 b
A%	8.8 ± 0.8 a	52.6 ± 1.3 b
a%	15.2 ± 0.5 a	79.6 ± 2.9 b

## Data Availability

Data are available from the authors upon request.

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
