# Peer review of "Plasticity of Plantago lanceolata L. in Adaptation to Extreme Environmental Conditions"

_ijms, 2023, doi:10.3390/ijms241713605_

Round 1
Reviewer 1 Report
Dear Authors,
Manuscript has potential, but it has some serious methodological flaws that needs to be adressed. Please see detailed comments in the attached document.

Author Response
Dear Editor,
thank you for your valuable comments on our MS Manuscript ID: ijms-2544062.
Title: Colonisation of Plantago lanceolata L. in adaptation to extreme environmental conditions
Authors: Zbigniew Miszalski, Paweł Kaszycki, Marta Śliwa-Cebula, Adriana Kaczmarczyk, Miron Gieniec, Paulina Supel, Andrzej Kornaś.
We checked all the references, and according to the Reviewers’ suggestions we also added new phrases. Now it seems they are relevant to the content of the MS.
Our answers and comments to the Reviewers are included below and any corrections have been highlighted in yellow in the text of MS.
Kind regards
Andrzej KornaÅ›
-----------------------------------------------------
Rev 1
Open Review
(x) I would not like to sign my review report
( ) I would like to sign my review report
Quality of English Language
( ) I am not qualified to assess the quality of English in this paper
( ) English very difficult to understand/incomprehensible
( ) Extensive editing of English language required
( ) Moderate editing of English language required
(x) Minor editing of English language required
( ) English language fine. No issues detected
|
Yes |
Can be improved |
Must be improved |
Not applicable |
|
|
Does the introduction provide sufficient background and include all relevant references? |
( ) |
( ) |
(x) |
( ) |
|
Are all the cited references relevant to the research? |
( ) |
(x) |
( ) |
( ) |
|
Is the research design appropriate? |
( ) |
( ) |
(x) |
( ) |
|
Are the methods adequately described? |
( ) |
( ) |
(x) |
( ) |
|
Are the results clearly presented? |
( ) |
(x) |
( ) |
( ) |
|
Are the conclusions supported by the results? |
( ) |
( ) |
(x) |
( ) |
Comments and Suggestions for Authors
Dear Authors,
Manuscript has potential, but it has some serious methodological flaws that needs to be adressed. Please see detailed comments in the attached document
Aim
Aim is not concise and clear enough. “To detect any possible change” is too broad to be a research aim. Moreover, study does not address the issue in a proper way. One way of addressing the issue that is presented as an aim is to use a common garden experiment with plants originating from dessert and forest ecosystem subjected to drought and regular watering treatments. Only based on such results it is possible to discuss about potential acclimatisation or adaptation processes.
The manuscript has serious flaws in M&M section.
- Experiment design is not adequately described and based on current description it is not possible to replicate the experiment.
- Microclimatic conditions of BD and YF locations are not provided therefore it is not possible to prove the difference in drought
- Measured leaf and root traits are not sufficient for estimation of plant drought adaptation. Traits that should also be included are: specific leaf area (SLA), leaf dry mass content (LDMC), leaf area (LA), fine and coarse root mass, above and belowground biomass ratio.
- Section 4.4 is not sufficiently explained
- Chl-a fluorescence measurements are highly variable depending on the daily conditions. In section 4.6 there is no information on when during the year measurements were performed. Nor what meteorological conditions where at the time of the measurements. Therefore, it is not possible to give any conclusions.
Conclusions
Due to flaws in methodology, obtained results are questionable and consequently results can not support conclusions.
For example, in lines 438-440 authors mention “big spreading root system” based solely on the visual assessment from the figure 1 D and E. Aside that, there is no other parameter that describes root growth at two locations.
Moreover, in lines 440-442 authors mention “large amount of proteins” that was not measured in this study. Amount of Rubisco is not mentioned neither in the M&M nor in the Results.
Also, in lines 443-444, it is given a sentence that confirms a well known fact about P. lanceolata, while in lines 444-450 authors give a conclusion that is not corroborated with results.
We thank the Reviewer 1 for insightful evaluation of our MS. Any corrections made have been highlighted in yellow in the text of MS.
We agree with your suggestion that the aim of our MS can be better formulated. Our aim was not “to use a common garden experiment with plants originating from desert and forest ecosystem subjected to drought and regular watering treatments” but rather to compare physiological parameters in plants exposed under natural condition to extreme environmental conditions and to check their plasticity. We modified formulation of the aim and the resulting conclusions of the study. (line 106-108, 110-111, 515-518)
In addition to the above we have also changed the title of our MS. In the title instead of “Colonisation of….” we use “Plasticity of…” to better understand the success of Plantago lanceolata in harsh sandy environment.
Ref. 1. and ref. 2. We completed the data describing the area of our investigations. (line 412-418)
The Błędów Desert in the southern part of Poland is formed of fluvioglacial sand (average thickness 40 m) with an admixture of gravel from the Pleistocene. The mean annual air temperature in the study area is 7 °C and the annual precipitation is 650 to 750 mm [45] (Gus and Drewnik, 2017). The area of the Błędów Desert is surrounded in the north and south by forest complexes. Hence, characteristic dune border ridges have been formed on the verge between the flat sandy surface and the forest [46] (Rahmonow et al. 2006). P. lanceolata usually grows on light well-aerated soils, grazing, sandy grasslands or black earth.
We added both publications to References.
Ref. 3. In our research we focus on characterizing photosynthetic machinery, as well anatomical parts responsible for water and nutrients transportation. As far as they were available we introduced some additional parameters, however, dry mass was not made available for us. Some additional information related to leaf structure can be withdrawn from Fig. 1 and cross-section shown in Fig. 2. We added a new phrase “In addition to this we have found that N/C ratio in plants from BD was higher than that obtained for YF locations (for lamina about 1.98 times higher and for leaf bundle 1.55 times higher – it was a significant increase).”(line 140-143)
Ref.4. We completed section 4.4. (line 464-471). Prior to analyses of carbon and nitrogen isotopes content, freeze-dried leaf material was homogenized in an agate mortar until a finegrained powder was obtained. Determination of the carbon isotope ratio, δ13C and δ15N, was performed with the continuous flow technique. The samples were weighed and then burned off in a furnace at 1020 °C. Next, they were transported within the helium stream through the Con-Flo IV Interface to the mass spectrometer to enable determination of the ratio of particular carbon isotope contents. The calculations were made employing internal standards USGS 40, USGS 41, and IAEA 600 [51] (Coplen et al. 2006).
Ref. 5. We agree with the Reviewer suggestion and we completed this information. The analyses were made under favorable atmospheric conditions, during warm and sunny days of July-August (maximum temperature of approximately 25 °C). (line 489-491)
Conclusions comments – MS
- 438-440 – figure 1D was typical example of at least 10 plants.
- 440-442 – In fact we did not measure protein content. This suggestion is based on well described observation reporting that RubisCO is the most abundant protein reaching in many plants 40-50% of all proteins. It can also be nitrogen reservoir.
- 443-444 – We removed this information
- 444-450 – According to your suggestion we changed this part of the text concerning biomass.
Thank you very much again for all your comments and suggestions.
Kind regards
Andrzej KornaÅ›
Reviewer 2 Report
The manuscript presents the results and conclusions of a field work of relative interest to understand the acclimatization capacity of Plantado lanceolata to different environmental situations, displaying certain phenotypic plasticity of some physiological and morphological characters.
The experiment is well designed but in my opinion it presents certain deficiencies in the approach of the material and methods that must be addressed to improve the clarity of the results and the discussion and conclusions.
Although the authors compare two clearly different environmental situations, it is essential to characterize them from the point of view of light, heat and water content in the soil in order to assess to what extent these parameters may be responsible for the differences found.
The lighting aspect of both locations is fundamental since it would allow to give meaning to the differences in Fv/Fm, for example. Forest plants could be in the shade, while desert plants could be in the sun. Likewise, it is necessary to carry out measurements over several consecutive days to differentiate damage by photoinhibition from the midday photoinhibition process. Normally, plants subjected to high solar radiation at noon present a photoinhibition mechanism that is protective, but plants usually recover at night, returning to normal photochemical efficiency values during the night and early hours of dawn. Only plants that suffer from photoinhibition damage will show a reduction in maximum photochemical efficiency (Fv/Fm) at dawn.
Authors should include statistical analysis on all LSD-type measured values to ensure whether the differences found are significant.
I am including a copy of the manuscript with some comments and corrections highlighted in yellow.

Author Response
Dear Editor,
thank you for your valuable comments on our MS Manuscript ID: ijms-2544062.
Title: Colonisation of Plantago lanceolata L. in adaptation to extreme environmental conditions
Authors: Zbigniew Miszalski, Paweł Kaszycki, Marta Śliwa-Cebula, Adriana Kaczmarczyk, Miron Gieniec, Paulina Supel, Andrzej Kornaś.
We checked all the references, and according to the Reviewers’ suggestions we also added new phrases. Now it seems they are relevant to the content of the MS.
Our answers and comments to the Reviewers are included below and any corrections have been highlighted in yellow in the text of MS.
Kind regards
Andrzej KornaÅ›
-----------------------------------------------------
Rev 2
Open Review
( ) I would not like to sign my review report
(x) I would like to sign my review report
Quality of English Language
(x) I am not qualified to assess the quality of English in this paper
( ) English very difficult to understand/incomprehensible
( ) Extensive editing of English language required
( ) Moderate editing of English language required
( ) Minor editing of English language required
( ) English language fine. No issues detected
|
Yes |
Can be improved |
Must be improved |
Not applicable |
|
|
Does the introduction provide sufficient background and include all relevant references? |
( ) |
(x) |
( ) |
( ) |
|
Are all the cited references relevant to the research? |
(x) |
( ) |
( ) |
( ) |
|
Is the research design appropriate? |
( ) |
( ) |
(x) |
( ) |
|
Are the methods adequately described? |
( ) |
( ) |
(x) |
( ) |
|
Are the results clearly presented? |
( ) |
( ) |
(x) |
( ) |
|
Are the conclusions supported by the results? |
( ) |
( ) |
(x) |
( ) |
Comments and Suggestions for Authors
The manuscript presents the results and conclusions of a field work of relative interest to understand the acclimatization capacity of Plantado lanceolata to different environmental situations, displaying certain phenotypic plasticity of some physiological and morphological characters.
The experiment is well designed but in my opinion it presents certain deficiencies in the approach of the material and methods that must be addressed to improve the clarity of the results and the discussion and conclusions.
We agree with your comment; we introduced new sentences. The corrections have been highlighted in yellow in the text of MS.
Although the authors compare two clearly different environmental situations, it is essential to characterize them from the point of view of light, heat and water content in the soil in order to assess to what extent these parameters may be responsible for the differences found.
We agree with your comment; we introduced new sentences. The corrections have been highlighted in yellow in the text of MS.
The lighting aspect of both locations is fundamental since it ould allow to give meaning to the differences in Fv/Fm, for example. Forest plants could be in the shade, while desert plants could be in the sun. Likewise, it is necessary to carry out measurements over several consecutive days to differentiate damage by photoinhibition from the midday photoinhibition process. Normally, plants subjected to high solar radiation at noon present a photoinhibition mechanism that is protective, but plants usually recover at night, returning to normal photochemical efficiency values during the night and early hours of dawn. Only plants that suffer from photoinhibition damage will show a reduction in maximum photochemical efficiency (Fv/Fm) at dawn.
Authors should include statistical analysis on all LSD-type measured values to ensure whether the differences found are significant.
I am including a copy of the manuscript with some comments and corrections highlighted in yellow.
We thank the Reviewer 2 for his insightful evaluation of our MS and very helpful comments in peer-review-31344320.v1.pdf. All corrections have been highlighted in yellow in the text of MS.
We introduced some data related to characterizing both investigated locations. We hope with this information it will be easier for potential reader to understand what are the causes responsible for development described differences in physiology of tested plants. (line 412-418)
Our main aim was to describe Plantago lanceolata acclimation to harsh conditions of desert area. Despite this we made our measurements during July-August during sunny weather and we checked acclimation of photosynthetic apparatus as well roots and veins and also development of arbuscular mycorrhiza in plants exposed to desert conditions. (line 106-108, 110-111, 489-491)
Thank you very much again for all your comments and suggestions.
Line 36. We replaced “adaptive” with “acclimation”.
Line 38-42. We added citation [3-5] to this part of the text.
Line 53. OK
Line 104. We replaced ”adaptation” with “acclimation”.
Line 111-114. The aim of this work was to compare physiological parameters in plants exposed in natural locations to extreme environmental conditions and to check their plasticity. We focus on the structure of veins as this is an important tissue responsible for transportation of water and nutrients.
Line 124-125. The difference is not significant. We introduced this to the text.
Line 129. We agree with this suggestion. The figures have been compiled to the same scale.
Line 132-135. The results show measurements of representative cross-sections from both sites. We introduced this information. (line 137-138)
Line 145-148. The microbiological frequency data are statistically significant whenever indicated in the text. Due to some uncertainties we have rewritten Chpt. 4.7 „Statistical analyses” to give straightforward information regarding every experimental set. For rhizospheric bacteria and yeast population assessment we have performed at least four replicate analyses and calculated mean values ± SD taking into consideration ALL the individual data. This means that for the case of the forest location where highly changeable frequencies of culturable bacteria were observed, the SD was relatively high and therefore the values given as 5.61±4.10 ∙106 CFU/g d.m., are reliable, solid and righteous. We conclude that the 9-fold difference in the colonization rate is statistically significant. In the text of chpt. 2.2., for clarity, we have added:
“... the rhizosphere of the plants growing at BD locations was colonized by a statistically significant higher number of bacteria...”. (line 153-155)
We believe that in simple direct microbial frequency analyses and comparison of two tested cases there is no need for more detailed statistical assessment than consideration of standard errors. Note that for given SD values we obtain the following range of results (bacterial frequency): for BD (desert): from 3.11 to 6.91 ∙107 CFU/g d.m., and for YF (forest): from 1.51 to 9.71 ∙106 CFU/g d.m., which shows clearly that in every case the difference in bacterial population is noted in favor of the desert rhizosphere.
For the second case (yeast) we conclude no difference in colonization rate. One of the values given in the text was misprinted by our unfortunate mistake and should stand as:
1.07±0.53 ∙105 CFU/g d.m. for YF and 1.50±0.22 ∙106 CFU/g d.m. for BD,
which can be clearly seen in Figure 3. We are thankful for drawing our attention to this particular case. It has now been corrected in the text of the manuscript.
Line 162-165. We believe that the statistical analyses based on the t-test bring enough support for the reliability of the mycorrhizal colonization data in comparative assessment to show significant differences between the forest- and desert-collected Plantago roots. For the two locations tested the ANOVA variance analysis is inappropriate (it requires at least 3 parameters) and thus an LSD testing was not possible (any post-hoc testing needs to be done employing ANOVA module), so the t-test was enough. The statistical method is described in the text of the manuscript (chptrs 4.3. and 4.7), and then further explained in the caption of Table 1.
We have also clarified the fact that n=90 refers to the number of root specimens examined (see Chpt. 4.7 „Statistical analyses”), not number of plant objects (impossible to collect in an environmentally protected area of BD). Still, having collected a number of complementary and supplementary data, in response to the Reviewer’s comment, below we show convincing statistical evaluation using LSD test (Fisher post-hoc), where it can be clearly seen that all the parameters tested differed significantly. To do this, however, we had to include yet another location as a third group of values, that is analyses of mycorrhiza of Plantago growing at the meadow conditions. We stress the fact, however, that in the paper content we only focused on comparison between the desert an forest conditions. Please, follow the statistics given here:
Tables with the results obtained after the Fisher LSD POST-HOC test based on the variance ANOVA analysis considering 3 different Plantago locations, i.e. Forest, Desert, and Meadow.
Consecutive tables were constructed each for the analyzed parameters A, a, M, m, respectively.
|
POST-HOC (Fisher LSD) |
|
|
|
|
The analyzed parameter: A |
Forest |
Meadow |
Desert |
|
Differences of means |
|
|
|
|
Forest |
|
55.216667 |
43.772222 |
|
Meadow |
55.216667 |
|
11.444444 |
|
Desert |
43.772222 |
11.444444 |
|
|
NIR |
Forest |
Meadow |
Desert |
|
Forest |
|
2.31898 |
2.31898 |
|
Meadow |
2.31898 |
|
2.31898 |
|
Desert |
2.31898 |
2.31898 |
|
|
t statistics |
Forest |
Meadow |
Desert |
|
Forest |
|
58.262782 |
46.186987 |
|
Meadow |
58.262782 |
|
12.075796 |
|
Desert |
46.186987 |
12.075796 |
|
|
p value |
Forest |
Meadow |
Desert |
|
Forest |
|
<0.000001 |
<0.000001 |
|
Meadow |
<0.000001 |
|
0.00002 |
|
Desert |
<0.000001 |
0.00002 |
|
|
Homogeneous groups |
Forest(a) |
Meadow(c) |
Desert(b) |
|
A |
* |
|
|
|
B |
|
|
* |
|
C |
|
* |
|
|
POST-HOC (Fisher LSD) |
|
|
|
|
The analyzed parameter: a |
Forest |
Meadow |
Desert |
|
Differences of means |
|
|
|
|
Forest |
|
58.692559 |
64.401046 |
|
Meadow |
58.692559 |
|
5.708487 |
|
Desert |
64.401046 |
5.708487 |
|
|
NIR |
Forest |
Meadow |
Desert |
|
Forest |
|
3.685053 |
3.685053 |
|
Meadow |
3.685053 |
|
3.685053 |
|
Desert |
3.685053 |
3.685053 |
|
|
t statistics |
Forest |
Meadow |
Desert |
|
Forest |
|
38.972421 |
42.762911 |
|
Meadow |
38.972421 |
|
3.79049 |
|
Desert |
42.762911 |
3.79049 |
|
|
p value |
Forest |
Meadow |
Desert |
|
Forest |
|
<0.000001 |
<0.000001 |
|
Meadow |
<0.000001 |
|
0.009068 |
|
Desert |
<0.000001 |
0.009068 |
|
|
Homogeneous groups |
Forest(a) |
Meadow(b) |
Desert(c) |
|
A |
* |
|
|
|
B |
|
* |
|
|
C |
|
|
* |
|
POST-HOC (Fisher LSD) |
|
|
|
|
The analyzed parameter: m |
Forest |
Meadow |
Desert |
|
Differences of means |
|
|
|
|
Forest |
|
28.722222 |
8.166667 |
|
Meadow |
28.722222 |
|
20.555556 |
|
Desert |
8.166667 |
20.555556 |
|
|
NIR |
Forest |
Meadow |
Desert |
|
Forest |
|
5.422832 |
5.422832 |
|
Meadow |
5.422832 |
|
5.422832 |
|
Desert |
5.422832 |
5.422832 |
|
|
t statistics |
Forest |
Meadow |
Desert |
|
Forest |
|
12.960148 |
3.684994 |
|
Meadow |
12.960148 |
|
9.275154 |
|
Desert |
3.684994 |
9.275154 |
|
|
p value |
Forest |
Meadow |
Desert |
|
Forest |
|
0.000013 |
0.01027 |
|
Meadow |
0.000013 |
|
0.000089 |
|
Desert |
0.01027 |
0.000089 |
|
|
Homogeneous groups |
Forest(a) |
Meadow(c) |
Desert(b) |
|
A |
* |
|
|
|
B |
|
|
* |
|
C |
|
* |
|
|
POST-HOC (Fisher LSD) |
|
|
|
|
The analyzed parameter: M |
Forest |
Meadow |
Desert |
|
Differences of means |
|
|
|
|
Forest |
|
28.722222 |
8.166667 |
|
Meadow |
28.722222 |
|
20.555556 |
|
Desert |
8.166667 |
20.555556 |
|
|
NIR |
Forest |
Meadow |
Desert |
|
Forest |
|
5.422832 |
5.422832 |
|
Meadow |
5.422832 |
|
5.422832 |
|
Desert |
5.422832 |
5.422832 |
|
|
t statistics |
Forest |
Meadow |
Desert |
|
Forest |
|
12.960148 |
3.684994 |
|
Meadow |
12.960148 |
|
9.275154 |
|
Desert |
3.684994 |
9.275154 |
|
|
p value |
Forest |
Meadow |
Desert |
|
Forest |
|
0.000013 |
0.01027 |
|
Meadow |
0.000013 |
|
0.000089 |
|
Desert |
0.01027 |
0.000089 |
|
|
Homogeneous groups |
Forest(a) |
Meadow(c) |
Desert(b) |
|
A |
* |
|
|
|
B |
|
|
* |
|
C |
|
* |
|
In addition to the analyses of mycorrhizal colonization, to further support the observed arbuscular mycorrhiza promotion in the area of the arid desert sites, we have added a recent reference (Vasar et al. 2021, Microorganisms now: [29]) showing the importance of mutualistic arbuscular mycorrhizal fungi under drought-stressed conditions. The relevant text can now be seen in the Discussion section.
Line 201. In fact, as it is reported oxidative damages related to removing of PSBA (and also PSBC and PSBD) protein can be in many cases replaced with new PSBA protein, however in many cases this process takes longer and we have to deal with lasting oxidative damage. It is not mentioned about oxidative damages other than PSII.
Line 233-234. We added information on leaf morphology (leaf length/width ratio). (line 123-128).
Line 236-237. Our root observations were based on macroscopic analysis of many specimens, see photo:
Line 240-244 As for “counterintuitiveness”: we, too, were surprised by the obtained data but have to stress that, however unexpected, this is a result of reliable and replicative testing (see our comments above). In fact, bacteria proliferated only under canopies of plant vegetation, that is in the so-called “vegetation islands” where well-developed Plantago root systems provided favorable conditions for efficient colonization.
According to the Reviewer suggestion we have made an additional survey of the available literature, which, although scarce, supports our findings (the newly cited papers have all been added to the reference list).
Please note that in this study we focused on monitoring the microbial colonization rate by frequency determination of culturable strains (abundance assessment of bacteria and microscopic fungi). The detailed analysis of structures of microbial communities requires much more complex and thorough approach and here it was beyond the scope of the performed study. In turn, for the case of the desert-like bulk sandy soil, especially the one originating from the Błędów Desert, we have not carried out a systematic study on microbial population, focusing rather on the samples taken from the „vegetation islands” inhabited by Plantago as a pioneer plant. Still in several samplings done we have never determined microbial abundance higher than 10e4 which agrees with the data on culturable microbes reported for the Sekem desert soil in Egypt (new. ref. 25).
We have considered all the Reviewer’s comments for the former lines 240-245 in the paragraph partially rewritten and substantially extended in the Discussion section.
Line 308-312. In our opinion plants growing on sandy locations are not competing for N as the plants growing in the forest. These plants that are able to survive in desert conditions benefit with better supply of some nutrients. Such plants can be useful as pioneering plants in such areas. However we do not know the origin of N in both locations.
Penetration of nitrite and nitrate anions through the root epidermis and root tissue is better for lighter isotope 14N, however its transportation within vascular tissues and its reduction within roots and within leaves is more complicated. With 15N discrimination we see only the final effect of N abundance and supply/demand ratio.
Line 350 and 360. We corrected this.
Line 438. We removed this observation from Conclusion.
Thank you very much again for all your comments and suggestions.
Kind regards
Andrzej KornaÅ›

Round 2
Reviewer 2 Report
I want to thank the authors for the effort in answering the raised questions. Now I find the manuscript aceptable for publication. Nevertheless I want to tell the authors that an statistical analysis accompanying the mean values is very clarifier always.